# Proactive Decision Support for Glaucoma Treatment: Predicting Surgical Interventions with Clinically Available Data

**DOI:** 10.3390/bioengineering11020140

**Published:** 2024-01-30

**Authors:** Mark Christopher, Ruben Gonzalez, Justin Huynh, Evan Walker, Bharanidharan Radha Saseendrakumar, Christopher Bowd, Akram Belghith, Michael H. Goldbaum, Massimo A. Fazio, Christopher A. Girkin, Carlos Gustavo De Moraes, Jeffrey M. Liebmann, Robert N. Weinreb, Sally L. Baxter, Linda M. Zangwill

**Affiliations:** 1Hamilton Glaucoma Center and Division of Ophthalmology Informatics and Data Science, Shiley Eye Institute, Viterbi Family Department of Ophthalmology, University of California, San Diego, CA 92037, USA; mac157@health.ucsd.edu (M.C.);; 2Department of Ophthalmology and Vision Sciences, Heersink School of Medicine, University of Alabama at Birmingham, Birmingham, AL 35233, USA; 3Bernard and Shirlee Brown Glaucoma Research Laboratory, Department of Ophthalmology, Edward S. Harkness Eye Institute, Columbia University Medical Center, New York, NY 10032, USA

**Keywords:** glaucoma, glaucoma progression, glaucoma surgery, OCT, visual field, machine learning, deep learning

## Abstract

A longitudinal ophthalmic dataset was used to investigate multi-modal machine learning (ML) models incorporating patient demographics and history, clinical measurements, optical coherence tomography (OCT), and visual field (VF) testing in predicting glaucoma surgical interventions. The cohort included 369 patients who underwent glaucoma surgery and 592 patients who did not undergo surgery. The data types used for prediction included patient demographics, history of systemic conditions, medication history, ophthalmic measurements, 24-2 VF results, and thickness measurements from OCT imaging. The ML models were trained to predict surgical interventions and evaluated on independent data collected at a separate study site. The models were evaluated based on their ability to predict surgeries at varying lengths of time prior to surgical intervention. The highest performing predictions achieved an AUC of 0.93, 0.92, and 0.93 in predicting surgical intervention at 1 year, 2 years, and 3 years, respectively. The models were also able to achieve high sensitivity (0.89, 0.77, 0.86 at 1, 2, and 3 years, respectively) and specificity (0.85, 0.90, and 0.91 at 1, 2, and 3 years, respectively) at an 0.80 level of precision. The multi-modal models trained on a combination of data types predicted surgical interventions with high accuracy up to three years prior to surgery and could provide an important tool to predict the need for glaucoma intervention.

## 1. Introduction

Glaucoma is characterized by a progressive loss of vision and is the leading cause of irreversible blindness in the world [1]. Lowering intraocular pressure (IOP) is the only currently known effective way to slow disease progression, and available treatments focus on lowering IOP through medications, laser therapies, and/or surgical intervention [2,3,4]. However, it is difficult or impossible to predict the rate of glaucoma progression or identify which patients will require surgical intervention to prevent blindness [5]. The identification of patients with a high risk of progression will help to reduce the risk of vision loss and preserve patients’ quality of life.

Advances in artificial intelligence (AI) and machine learning (ML) over the past several years have provided tools that may help to address this need. AI-based approaches have had a large impact on many different prediction tasks across nearly all fields of medicine as well as numerous applications in ophthalmology and glaucoma [6,7,8,9]. Glaucoma care is data- and imaging-intensive. The current standard of care includes the collection of ophthalmic and systemic measurements, fundus and optical coherence tomography (OCT) imaging to assess the retinal structure, and visual field (VF) testing to evaluate visual function [10]. These sources of data have been exploited to build tools for glaucoma-related prediction tasks. Our group and others have employed AI techniques to detect disease, segment OCT images, and predict functions from structures, among other tasks [11,12,13,14,15,16]. These approaches, however, have largely relied on models that use a single data type. Multi-modal approaches that integrate the different types of data collected as part of routine clinical glaucoma management may improve the ability to identify glaucoma predictions. Moreover, they may address unmet needs, such as forecasting the need for surgical intervention in glaucoma.

There is increasing interest in multi-modal models that incorporate data types from multiple different sources into a single predictive model [17]. In the case of glaucoma predictions, potentially informative data sources can include not only ophthalmic measurements, OCT imaging, and VF testing but also patient demographics, medical history, and data regarding systemic conditions and medications [18,19]. The current study investigates the use of multi-modal models that incorporate each of these data types to predict the need for surgical intervention in varying time windows up to 3 years pre-intervention.

## 2. Materials and Methods

### 2.1. Datasets

The primary dataset was collected from a cohort of glaucoma participants recruited as part of two longitudinal glaucoma studies: the Diagnostic Innovations in Glaucoma Study (DIGS, clinicaltrials.gov identifier: NCT00221897) and the African Descent and Glaucoma Evaluation Study (ADAGES, clinicaltrials.gov identifier: NCT00221923) [20]. ADAGES is an ongoing, multicenter collaboration between four primary academic medical centers with high-volume glaucoma clinics and one high-volume private practice (delineated in this study as sites one through five for the purpose of preserving patient privacy). Details of the study design for these studies have been described previously [20]. In short, all participants provided written informed consent, and the institutional review boards at all sites approved the study methods. All methods adhere to the tenets of the Declaration of Helsinki and to the Health Insurance Portability and Accountability Act. Inclusion in the DIGS/ADAGES glaucoma cohorts required participants to meet the following criteria at study entry: 20/40 or better best corrected Snellen visual acuity and at least two consecutive reliable standard automated perimetry VF tests. For this analysis, glaucoma was defined as eyes having repeated abnormal VF results.

All clinical, demographic, VF testing, and OCT imaging data were stored in UCSD’s HIPAA-compliant Amazon Web Services (AWS)-based data management and analysis system, iDARE (intelligent Design for AI-Readiness in Eye Research).

### 2.2. Patient Exams and Interviews

An ophthalmological exam was completed at each study visit, which included measurements of IOP, central corneal thickness (CCT), spherical equivalent (SE), and axial length (AL). Patient interviews were also conducted to collect self-reported demographic information (age, sex, race, and ethnicity), systemic medical conditions, history of non-ocular medications, and history of ocular medications and interventions. Medications were mapped to the anatomical therapeutic chemical (ATC) drug classification defined by the World Health Organization using RxNav [21,22]. Complete details on the ADAGES examination and interview procedures have been described previously [20]. Both medical record data and self-reported history of interventions were used as the basis for the surgery ground truth used in the model’s training and testing. Self-reported data were used because clinical records were not available for some of the patients in the DIGS/ADAGES cohorts considered here.

### 2.3. OCT Imaging

OCT imaging consisted of Spectralis (Heidelberg Engineering GmbH, Heidelberg, Germany) circle scans containing single B-scans comprised of 1536 A-scans, each captured in a circular region surrounding the optic nerve head (ONH). The retinal nerve fiber layer (RNFL) was segmented using built-in software provided by the device manufacturer. The segmentation resulted in global and sectoral mean RNFL thickness measurements. The mean RNFL included global, temporal, temporal-superior, temporal-inferior, nasal, nasal-superior, and nasal-inferior. The OCT images and segmentations were evaluated for quality by the UC San Diego Imaging Data Evaluation and Analysis (IDEA) Reading Center according to standard protocols [20]. The OCT images with poor signal quality, imaging artifacts, and/or segmentation errors that could not be manually corrected were excluded from further analysis.

### 2.4. Visual Field Testing

VF testing consisted of 24-2 testing using the Swedish interactive thresholding algorithm (SITA standard, Humphrey Field Analyzer II; Carl Zeiss Meditec, Inc., Jena, Germany). VF results with more than 33% fixation losses, 33% false-negative errors, or 33% false-positive errors were excluded. The VF results were processed and evaluated to assess their quality according to standard protocols by the University of California San Diego Visual Field Assessment Center [20]. For inclusion in the models, the mean deviation (MD), pattern standard deviation (PSD), visual field index (VFI), and individual test point pattern deviation (PD) measurements were extracted.

### 2.5. Multi-Modal Models

The model input was constructed by combining the data sources described above into a final set of inputs. This set consisted of patient demographics (age, sex, and race), systemic medical conditions (e.g., hypertension, diabetes, and cancer), history of non-ocular and ocular medications, ophthalmic measurements (IOP, CCT, SE, and AL) along with 24-2 VF results (MD, PSD, VFI, and individual test point PD), and RNFL measurements (mean global and sectoral thickness). A complete list of model inputs and outputs is provided in the Appendix A. For this analysis, the model input consisted of quantitative (e.g., RNFL thickness, VF MD, age, IOP, etc.), categorical (e.g., self-reported race), and binary (e.g., gender and presence of systemic conditions) variables derived from the multiple modalities described above. Eyes missing VF, OCT, and IOP data were excluded from the final datasets in our study. The primary outcome of interest was whether the patients progressed to requiring glaucoma surgical intervention. The ground truth labels for the outcome (glaucoma surgery or no surgery) were identified based on the self-reported data collected during the patients’ exams. For this analysis, any patients with incisional, laser, or minimally invasive glaucoma surgery (MIGS) procedures were included in the surgery group, as these procedures would still typically represent advancing disease. This is also consistent with the methodology employed in prior studies examining glaucoma surgery as a proxy for disease progression [13,14,18]. Figure 1 provides surgery and non-surgery examples.

The longitudinal aspect of our dataset was used to simulate predicting surgeries at various lengths of time in the future. To this end, we defined a time horizon for our models. The time horizon represents the period of time prior to the surgery date during which no data is used as input to the model for training or testing. This is important for establishing applicability to future deployments of these models in real-world clinical environments. For example, as a clinician, one may want to understand a specific patient’s risk of progressing to needing surgery within the next year. With this framework in mind, when training models with existing data, it would not be sensible to include all data leading up to surgery. Therefore, a model with a time horizon of 12 months would be trained and tested only on input data that was collected at least 12 months prior to the surgery date.

To quantify how the accuracy of our models changed as we predicted surgeries at various lengths of time in the future, we trained and evaluated models on several time horizons, including 3 months, 1 year, 2 years, and 3 years. For each patient who did not progress to surgery, one study visit was randomly selected as a “pseudo surgery” date (to establish an analogous observation period) and was used to identify the data eligible for each time horizon. The actual input data to the models consisted of a single measurement for each feature selected from the most recently collected eligible data (based on the time horizon).

Several different model types were evaluated, including logistic regression, random forests, gradient-boosting machines (GBMs), gradient-boosted decision trees (XGBoost), and custom deep neural networks (DNNs) [23,24,25,26]. In all cases, the model parameters were selected via empirical testing on the training data.

### 2.6. Model Evaluation

The data were partitioned into training and testing datasets by selecting data collected at most sites to serve as a training/validation dataset (sites one to four) and one study site (site five) to serve as an external test dataset. This approach ensured that the testing data was collected from an independent, geographically separated population. The training dataset was further split (90/10, according to participant) into training and validation subsets. Models of different types with a range of parameters were trained on the training subset and evaluated on the validation subset. The model type with the best overall performance on the validation subset was selected for evaluation on the testing dataset. The models were evaluated using the area under the receiver operating characteristic (AUC) and precision-recall curves, as well as precision, sensitivity, and specificity. Using these metrics, the models were evaluated on the entire testing dataset and on the subsets stratified by age, self-reported race, and disease severity to estimate the impact of these factors on the model’s performance. When evaluating the models, we aimed to optimize the precision (also known as the positive predictive value), which is the proportion of patients who actually progressed to glaucoma surgery among those who were predicted to progress. This was in line with the primary goal of the future clinical deployment of this model. Subsequently, decision thresholds were tuned so that each model achieved a precision score of at least 0.80 while also maximizing the sensitivity. We then also evaluated how the model performance metrics varied at various thresholds for precision (from 0.75 to 0.95 in 0.05 increments).

To enhance model explainability, the decision-making process employed by the best-performing model was explored using Shapley additive explanations (SHAPs) [27]. SHAP is a game theory-based approach to measure the impact of input features on model output. Using this approach, we quantified the contribution of each input to predictions regarding the need for glaucoma surgery.

## 3. Results

A summary of the surgery and non-surgery cohorts used to train and test the models to predict glaucoma surgical intervention is provided in Table 1. In both the training and testing data, the surgery group was, on average older (68.4 years vs. 63.2 years in the training data, *p* < 0.001), had worse 24-2 MD (−7.17 dB vs. −1.40 dB in the training data, *p* < 0.001), and had a larger proportion of Black/African American patients (51.8% vs. 41.3% in the training data, *p* < 0.001). 

The best-performing surgical intervention prediction model was a GBM. It achieved an AUC (95% CI) of 0.94 (0.91–0.96) at the 3-month time horizon, incorporating all available data preceding surgery, 0.93 (0.90–0.95) at 1 year prior to surgery, 0.92 (0.89–0.95) at 2 years prior to surgery, and 0.93 (0.89–0.97) at 3 years prior to surgery. The decision thresholds were tuned so that each model achieved a precision score of at least 0.80 while also maximizing the sensitivity. For models with the 0.80 precision threshold, the specificity increased with longer time horizons (0.81, 0.85, 0.90, and 0.91 at 3 months, 1 year, 2 years, and 3 years, respectively), while sensitivity/recall was at a maximum at 3 months (0.93, 0.89, 0.77, and 0.86 at 3 months, 1 year, 2 years, and 3 years, respectively). The full results for this model are shown in Table 2 and illustrated as ROCs and precision-recall curves in Figure 2. The results for all the models are summarized in Appendix A.

The best-performing model (GBM) was also evaluated as a function of disease severity, age, and self-reported race using stratified analyses (Table 3). Comparing patients with 24-2 MD > −6.0 dB to those ≤−6.0 dB, the model achieved AUCs of 0.89 vs. 0.88 at 3 months, 0.90 vs. 0.78 at 1 year, 0.87 vs. 0.86 at 2 years, and 0.88 vs. 0.96 at 3 years. With respect to age, the model had similar AUCs for patients below and above 60 years old at all time horizons (AUCs of 0.94 vs. 0.93 at 3 months, 0.93 vs. 0.91 at 1 year, 0.92 vs. 0.92 at 2 years, and 0.94 vs. 0.92 at 3 years). With respect to self-reported race, only the Black/African American and White groups had sufficient patients to perform the analyses. Similar model performance was found in Black/African American vs. White patients; the AUCs were 0.93 vs. 0.94 at 3 months, 0.91 vs. 0.94 at 1 year, 0.92 vs. 0.94 at 2 years, and 0.94 vs. 0.93 at 3 years. Of note, no statistically significant differences at the 0.05 level were observed in the model performance according to disease severity, age, or race. 

The SHAP analysis of the model predictions revealed the features with the greatest impact on the predictions at each time horizon (Figure 3). Some particularly impactful features were common among all the time horizons (age, gender, self-reported race, AL, CCT, IOP, MD, VFI, and PSD). At shorter time horizons, the RNFL measurements were more important. At longer time horizons, the list of the most impactful features was dominated by the VF measurements. The self-reported patient conditions and medications did not appear among the most impactful features.

## 4. Discussion

### 4.1. Model Performance

In this study, we developed ML models that achieved high accuracy in predicting surgical intervention in glaucoma up to 3 years prior to the intervention. Even at 3 years prior to surgery (meaning no data were used within 3 years preceding the surgical intervention for training the model), the model achieved a high AUC (0.93) and sensitivity (0.86) as well as high specificity (0.91) and precision (0.80). Accurate methods to identify patients who are at high risk of progression and need glaucoma interventions, like those presented here, are a critical need in glaucoma. For patients, the early identification of risk may help to inform downstream interventions that can preserve vision and quality of life. For clinicians, forecasting patients who are at a high risk of glaucoma progression can help to inform which patients need closer monitoring and enable more efficient allocation of limited clinical time and resources.

Models were evaluated at several time horizons (i.e., predictions at different time periods prior to an intervention) spanning from 3 months up to 3 years prior to the surgery date. This is clinically relevant, as we plan to implement these models for point-of-care decision support in a prospective fashion. Clinicians would be interested in predicting a specific patient’s risk of requiring glaucoma surgery in the future and would be limited to whatever data may be currently available to them. Thus, while developing these models with retrospective data, we decided to censor data preceding surgery of varying lengths to simulate the prospective clinical implementation scenario. Developing models to predict the risk of progression to surgery up to 3 years in advance represents an improvement over prior studies that have examined shorter time windows for prediction, such as 6 months [13]. Compared to 3 months, longer time horizons performed better than expected. While the 3-month predictions did achieve the highest sensitivity, it was comparable to or worse than the other time horizons with respect to the AUC, precision, and specificity. Part of the similarity in performance across time horizons may be due to the availability of data during the corresponding time periods. Different time horizon models were restricted in what data they had access to, but the restrictions only limited model access to data nearer to surgery. Shorter horizon models did have access to older data and could use it if newer measurements were not available. For instance, the 1-year model could not use any time collected within the year prior to surgery but could use older data if needed. For a specified patient, if all the data for a particular measurement had been collected more than 3 years prior to surgery, all the time horizon models would have access to that data. In addition, some variables are stable over time and would be the same for all time horizons (e.g., some patient demographics). This means that, in some cases, models at different time horizons were relying on similar input to identify their predictions, possibly leading to greater-than-expected similarity. 

In estimating the performance of the models, we relied on the multi-center data collected in the DIGS/ADAGES dataset. Patient recruitment and data collection were performed at five geographically separated locations across the U.S. from demographically diverse populations. This provided the opportunity to withhold data from one study site to use as an independent test set to better estimate the generalizability of the models. The lack of external validation or test datasets is a commonly known challenge in the reproducibility of AI/ML models, so the evaluation of model performance in a completely independent set of patients is an important advancement over some prior clinical prediction models that largely relied on internal validation alone [28,29,30]. The diversity of our datasets also allowed for estimates of model performance across patient subgroups using stratified analyses based on disease severity, age, and self-reported race. There were some differences in model performance based on severity. The model performed better in patients with MD > −6.0 dB at 1 year and better in patients with MD ≤ −6.0 dB at 3 years, although these differences were not statistically significant. With respect to age and self-reported race, the models performed comparably at all the time horizons in both the over and under 60 years patient groups and the Black/African American and White patient groups. This is an important finding, as several recent studies have found other clinical AI models demonstrating inferior performance in minority populations, potentially creating a source of algorithmic bias with the potential to exacerbate existing health disparities [31,32,33]. Given that Blacks carry a disproportionate burden of glaucoma and face existing disparities in care, the high performance of our models in this population is important in the context of health equity [34,35]. Regardless of the stratified group, the model AUC remained high (≥0.88, with the most substantially higher), suggesting that the models could maintain performance across a wide variety of patient subgroups. The validation of these models in additional datasets, especially in real-world clinical data from diverse patient populations, is a critical next step toward deploying these models for clinical use.

### 4.2. Feature Importance

The Shapley analysis that was used to interrogate model decision-making may help explain the model performance at different time horizons. The incorporation of this analysis is also helpful for enhancing the explainability of the models, which is important considering that ML models have frequently been criticized for their “black box” nature [36]. Across all time horizons, several features had a moderate-to-large impact on the model predictions, including age, gender, self-reported race, axial length, CCT, IOP, MD, VFI, and PSD. Model reliance on the features that were relatively stable in impact across the time horizons may help to explain the similarity in performance at different time horizons. Other features, however, had large jumps in their impact at later time horizons. In particular, the impact of OCT-based RNFL thickness measurements had a larger impact during shorter time horizons. At longer time horizons, the relative importance of RNFL measurements decreased compared to the other features. Given the ubiquity of OCT in glaucoma management, this decrease in informativity at longer time horizons warrants further study. Overall, the Shapley analysis provided information about relative feature importance that coincided with known clinical features of glaucoma, instilling confidence in the models.

### 4.3. Limitations and Future Directions

There are limitations to the current study. First, there are differences in the rates of surgical interventions across clinicians and departments that could impact model performance. Across the three sites with the largest number of patients, the rates of glaucoma surgery cases were 24.7% (site one), 53.6% (site two), and 40.0% (site five). These are comparable to prior studies from both academic medical centers and nationwide cohorts such as the All of Us Research Program and the IRIS registry, which also demonstrate a high level of variability in the rates of glaucoma surgery [13,14,37,38]. The range of rates could be the results of differences in the patient populations, differences in decision-making regarding glaucoma interventions and local practice variations, patients refusing or postponing surgery, or a combination of these factors. Incorporating training and testing data from additional sites could help in training and evaluating models that are more robust to these differences. This additional data could also be useful in evaluating model performance across types of surgical interventions (laser, incisional, and MIGS). The current dataset has a limited number of subtypes (especially MIGS), limiting our ability to evaluate performance across subtypes. In addition, we limited our analyses to patients with available VF, OCT, and IOP data. However, missing data is often an issue in real-world clinical settings and may limit the applicability of our models to those settings. Future work will include developing models that can handle missingness without a large loss of performance, as this is an important consideration for clinical adoption [39]. Finally, the surgery ground truth was based on both the clinical record and self-reported data because clinical records were not available for many patients in the cohort. While this approach allowed us to include a larger number of patients in the analysis, there could be issues with accuracy in patient self-reporting. Future work will incorporate data from clinical records to determine surgical intervention ground truth.

There are several possible future directions to build on the current study, which include (1) incorporating raw OCT and fundus photography data into models, (2) extending the current methods to better take advantage of longitudinal data, and (3) working to incorporate models into clinical workflows. The models reported in the current study only used summary metrics (mean RNFL thickness values) and did not take advantage of the information represented by the raw OCT image data. Incorporating this data (as well as fundus photos) could help to improve performance even further. Additionally, the current study identified predictions largely based on measurements collected at single visits. Extending our methods to use serial data from clinical, imaging, and VF testing visits could also improve accuracy or extend the timespan of surgical intervention prediction. A variety of machine and deep learning methods exist (e.g., recurrent neural networks, transformers, etc.) that explicitly model the longitudinal aspects of our datasets [40,41]. Our longer-term goal is to incorporate these predictive models into clinical settings so that they can be utilized at the point of care to help identify high-risk patients and inform downstream impacts on patient care. To this end, future work will also focus on validating methods in real-world clinical data and developing computational infrastructure to provide clinicians with real-time predictions to support their clinical decision-making. 

## 5. Conclusions

In summary, our ML estimates achieved high accuracy in predicting surgical interventions in glaucoma up to 3 years in advance. The model accuracy was consistently high across age and racial subgroups in the test dataset. These results show that multi-modal ML approaches can achieve high accuracy in a critical glaucoma prediction task and suggest the potential for a large impact on patient care.

## Figures and Tables

**Figure 1 bioengineering-11-00140-f001:**
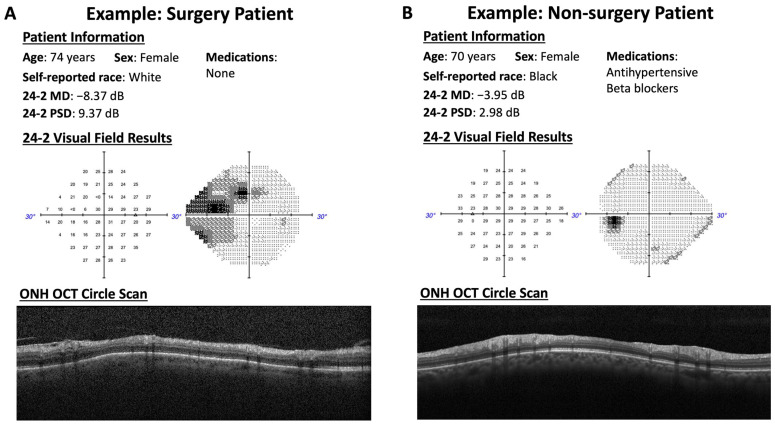
Example of a patient who progressed to needing glaucoma surgery and (**A**) another patient who did not end up needing surgery. (**B**) Patient demographics, medication history, visual field results, and optic nerve head circle scans.

**Figure 2 bioengineering-11-00140-f002:**
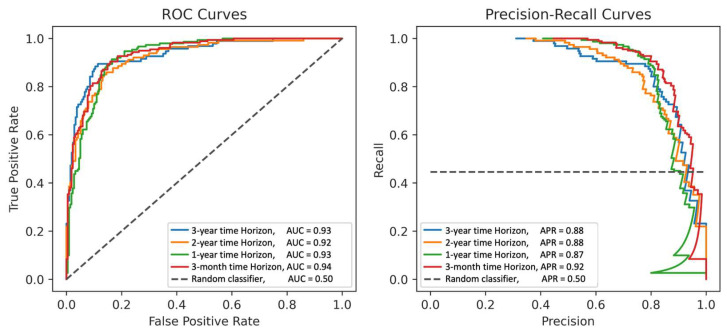
Receiver operating characteristics and precision-recall curves of the best-performing model (GBM) at each time horizon.

**Figure 3 bioengineering-11-00140-f003:**
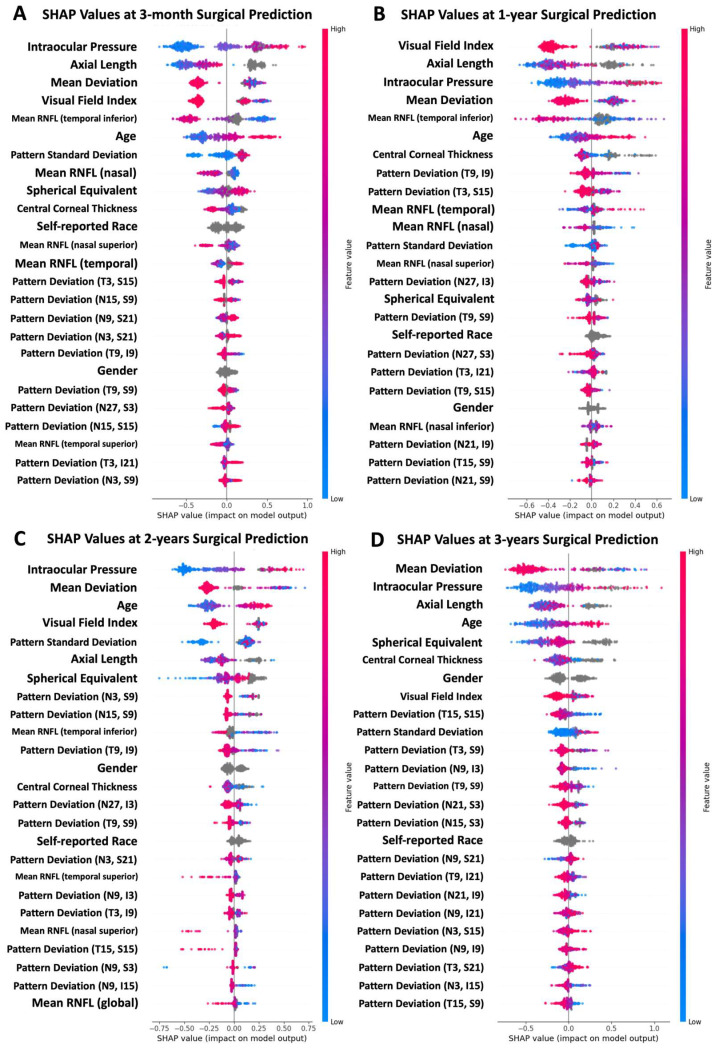
Shapley analysis of the features associated with the greatest impact on predictions for the best-performing model trained with all data leading to surgery. (**A**) The best-performing model trained with data up to 1 year prior to surgery, (**B**) up to 2 years prior to surgery, and (**C**) up to 3 years prior to surgery. (**D**) In these plots, each point corresponds to a single patient, and the color (blue to pink) indicates the normalized value of the indicated feature for that patient. The x-axis represents the SHAP value (the contribution of the indicated feature to the model prediction for each patient). The positive values pushed the model toward a surgery prediction, while the negative values pushed the model toward a non-surgery prediction. The features with all points clustered near zero had relatively little impact on the model decisions, while the features with higher SHAP values had a larger impact.

**Table 1 bioengineering-11-00140-t001:** A summary of the testing and training datasets. The training data consisted of data collected at sites one through four, while site five data was held out as an independent testing dataset.

	Training		Testing	
	Glaucoma Surgery	No Surgery	Glaucoma Surgery	No Surgery
**Participants/eyes (n, %)**	419/610 (45.8%)	496/830 (54.2%)	137/178 (51.9%)	127/221 (48.1%)
Age (years, 95% CI)	68.4 (67.5–69.4)	63.2 (62.3–64.2)	66.3 (64.5–68.0)	59.8 (58.0–61.6)
24-2 MD (dB, 95% CI)	−7.17 (−7.70–−6.63)	−1.40 (−1.85–−0.94)	−8.39 (−9.74–−7.05)	−1.19 (−2.18–−0.20)
Mean RNFL thickness (μm, 95% CI)	75.2 (72.2–78.1)	86.6 (84.8–88.3)	73.3 (68.1–78.5)	84.4 (79.6–89.2)
Sex (n, % female)	217 (51.8%)	306 (61.7%)	91 (66.4%)	71 (55.9%)
**Self-reported race (n, %)**				
Black/African American	217 (51.8%)	205 (41.3%)	90 (65.7%)	67 (52.8%)
White	194 (46.3%)	253 (51.0%)	47 (34.3%)	60 (47.2%)
Other/not reported	8 (1.9%)	38 (7.7%)	0 (0.0%)	0 (0.0%)
**Surgery type**				
Incisional	170 (27.9%)	-	67 (37.6%)	-
MIGS	1 (0.2%)	-	6 (3.4%)	-
Laser	439 (72.0%)	-	105 (59.0%)	-
None	0 (0.0%)	830 (100.0%)	0 (0.0%)	221 (100.0%)

**Table 2 bioengineering-11-00140-t002:** Performance of the best-performing model (GBM) in predicting glaucoma surgical interventions at time horizons up to 3 years. The models used patient demographics, ophthalmic measurements, VF results, OCT measurements, and self-reported history of systemic conditions and medications to predict surgical interventions. Multiple tuning thresholds were used for evaluating the model performance metrics at varying levels of precision.

Precision at Time Horizons	AUC	Precision	Recall/Sensitivity	Specificity
3 months	0.94 (0.91–0.96)			
0.75 precision		0.75 (0.71–0.79)	0.94 (0.91–0.98)	0.75 (0.69–0.80)
0.80 precision		0.80 (0.76–0.85)	0.93 (0.89–0.96)	0.81 (0.76–0.86)
0.85 precision		0.85 (0.80–0.90)	0.83 (0.78–0.88)	0.88 (0.84–0.92)
0.90 precision		0.90 (0.85–0.95)	0.68 (0.61–0.75)	0.94 (0.91–0.97)
0.95 precision		0.95 (0.91–0.99)	0.56 (0.49–0.63)	0.98 (0.96–1.00)
1 year	0.93 (0.90–0.95)			
0.75 precision		0.75 (0.71–0.80)	0.95 (0.91–0.98)	0.79 (0.73–0.84)
0.80 precision		0.80 (0.75–0.85)	0.89 (0.84–0.93)	0.85 (0.80–0.90)
0.85 precision		0.85 (0.80–0.91)	0.66 (0.59–0.74)	0.92 (0.89–0.95)
0.90 precision		0.91 (0.84–0.97)	0.45 (0.37–0.53)	0.97 (0.94–0.99)
0.95 precision		0.96 (0.89–1.00)	0.30 (0.22–0.38)	0.99 (0.98–1.00)
2 years	0.92 (0.89–0.95)			
0.75 precision		0.75 (0.70–0.82)	0.86 (0.79–0.92)	0.85 (0.80–0.90)
0.80 precision		0.80 (0.74–0.86)	0.77 (0.69–0.85)	0.90 (0.86–0.94)
0.85 precision		0.85 (0.78–0.91)	0.70 (0.61–0.78)	0.94 (0.90–0.97)
0.90 precision		0.90 (0.84–0.97)	0.58 (0.48–0.67)	0.97 (0.94–0.99)
0.95 precision		0.96 (0.90–1.00)	0.39 (0.30–0.47)	0.99 (0.98–1.00)
3 years	0.93 (0.89–0.97)			
0.75 precision		0.75 (0.69–0.82)	0.90 (0.83–0.95)	0.87 (0.82–0.92)
0.80 precision		0.80 (0.74–0.88)	0.86 (0.79–0.93)	0.91 (0.87–0.94)
0.85 precision		0.85 (0.78–0.92)	0.75 (0.66–0.83)	0.94 (0.91–0.97)
0.90 precision		0.90 (0.84–0.97)	0.67 (0.58–0.77)	0.97 (0.94–0.99)
0.95 precision		0.97 (0.90–1.00)	0.33 (0.23–0.43)	0.99 (0.99–1.00)

**Table 3 bioengineering-11-00140-t003:** AUC of the best-performing model (GBM) stratified by age, self-reported race, and glaucoma severity at each time horizon.

	3 Months	1 Year	2 Years	3 Years
Glaucoma severity				
MD > −6.0 dB (n = 153)	0.89 (0.84–0.93)	0.90 (0.86–0.94)	0.87 (0.79–0.92)	0.88 (0.79–0.94)
MD ≤ −6.0 dB (n = 50)	0.88 (0.72–0.97)	0.78 (0.58–0.92)	0.86 (0.70–0.96)	0.96 (0.84–1.00)
Age				
>60 years (n = 113)	0.93 (0.88–0.96)	0.91 (0.86–0.95)	0.92 (0.86–0.96)	0.92 (0.86–0.96)
≤60 years (n = 107)	0.94 (0.89–0.97)	0.93 (0.89–0.96)	0.92 (0.85–0.96)	0.94 (0.87–0.99)
Self-reported race				
Black/African American (n = 134)	0.93 (0.89–0.96)	0.91 (0.87–0.95)	0.92 (0.86–0.96)	0.94 (0.88–0.97)
White (n = 86)	0.94 (0.90–0.97)	0.94 (0.90–0.97)	0.94 (0.88–0.97)	0.93 (0.85–0.98)

## Data Availability

The data presented in this study are available on reasonable request from the corresponding author. The data are not publicly available due to potential privacy issues.

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
