# Peer review of "Proactive Decision Support for Glaucoma Treatment: Predicting Surgical Interventions with Clinically Available Data"

_bioengineering, 2024, doi:10.3390/bioengineering11020140_

Round 1
Reviewer 1 Report
Comments and Suggestions for Authors
Regarding the manuscript titled "Predicting Surgical Interventions for Glaucoma with Clinical Available Data," the authors utilized a longitudinal ophthalmic dataset to develop multi-modal machine learning models that effectively predicted glaucoma surgical interventions. These models incorporated diverse data types, including patient demographics, clinical measurements, optical coherence tomography (OCT), and visual field testing. Remarkably, the highest-performing model, a gradient boosting machine (GBM), achieved impressive AUC values of 0.93, 0.92, and 0.93 when predicting surgical interventions at 1, 2, and 3 years respectively. This demonstrates the accurate prediction of glaucoma intervention needs up to three years in advance, offering a valuable tool for guiding clinical decision-making.
We would like to emphasize that this manuscript exhibits objective innovation and holds clinical significance. Furthermore, the content organization is commendable, presenting well-structured and coherent information. Although the study methodology is iterative, it has been executed flawlessly. However, to further enhance the quality of this manuscript and ensure its suitability for publication, we suggest considering the following revisions:
1. It is advisable to avoid using a generic and commonplace title for this study. We propose a more fitting title: "Proactive Decision Support for Glaucoma Treatment: Predicting Surgical Interventions with Clinically Available Data." However, the final decision regarding the title rests with the authors, and our suggestion is merely a recommendation.
2. We strongly recommend discussing the applications of machine learning in healthcare within the introduction and providing a justification for the importance of artificial intelligence and machine learning in healthcare. The following articles can be referenced to enrich the manuscript and provide readers with relevant resources on artificial intelligence:
- "Toward artificial intelligence (AI) applications in the determination of COVID-19 infection severity: considering AI as a disease control strategy in future pandemics."
- "AI-driven malaria diagnosis: developing a robust model for accurate detection and classification of malaria parasites."
- "A hybrid particle swarm and neural network approach for detection of prostate cancer from benign hyperplasia of the prostate."
- "Clinical decision support system for early detection of prostate cancer from benign hyperplasia of the prostate."
Incorporating these references will not only enhance the manuscript's quality but also provide valuable insights into the application of artificial intelligence in healthcare.
3. Pay careful attention to the metrics of the top model in your study when summarizing the findings in the abstract. Strive to provide a comprehensive and succinct overview of all the results.
4. As this is a novel topic, it would be beneficial to include a section dedicated to future suggestions within the study and the conclusion. In this section, you can offer recommendations to other specialists, encompassing both technical aspects and clinical processes related to this study. This addition will enhance the readability and practicality of your research.
In conclusion, we acknowledge the well-written nature of this study; however, minor revisions are necessary. We kindly request that you meticulously address these revisions to enable us to recommend the manuscript for publication to the journal's editor.
Reviewer 2 Report
Comments and Suggestions for Authors
This study aimed to build machine learning models to predict surgical intervention in 3 months, 1 year, 2 years, and 3 years for glaucoma patients (surgery: 369; no-surgery 592 glaucoma), based on data collected from the current standard of care, including patient demographics, medical conditions, history of non-ocular and ocular medications, ophthalmic measurements, 24-2 VF results, and RNFL measurements. The gradient boosting machine (GBM) achieved impressive AUCs of above 0.92. SHAP was used for model interpretation, identifying important features for the prediction. However, there are several issues that need to be addressed in this study.
1. This study only used patient data at a single visit to predict surgical intervention in 3 months, 1 year, 2 years, and 3 years, but the fact that only a single visit was used was not revealed until in the Discussion (Line 341-342), which I found very misleading, especially when the longitudinality of the data was emphasized several times in the text. The authors are suggested to make this clear in Section 2.
2. The concept of multi-modal learning in ML refers to using different data types, e.g., image, text, numerical values, etc. as model inputs. The final set of input features of this work (Lines 112-116) contains only numerical values. Therefore, the features might be extracted from multiple data modalities, but the models trained on these features should not be called “multi-modal” models; otherwise, it is very misleading the readers again.
3. The ground truth labels (glaucoma surgery or no surgery), as the authors stated in Lines 120-122, were identified based on the “self-reported data” collected during patient exams. Why was such important information for the whole research target collected from patient’s self-reported data, instead of clinical records?
4. Amongst three types of surgeries under consideration, Laser was dominant, followed by incisional, but MIGS only had a few cases in both the training and test sets (Table 1). It would be interesting to see how the best model performs on these subtypes.
Comments on the Quality of English LanguageOK
Reviewer 3 Report
Comments and Suggestions for Authors
Thanks for offering the opportunity to review this paper. The authors tried to investigate multi-modal ML models incorporating patient demographics and history, clinical measurements, OCT) and VF testing in predicting glaucoma surgical interventions.
Overall comment: A nice work.
Minor revision suggestions:
1. There are some grammatical errors or clerical errors in the manuscript
2. The clarity of Figure 3 needs to be improved
3. The discussion part is somewhat too long and tedious
Comments on the Quality of English LanguageMinor editing of English language required
Round 2
Reviewer 2 Report
Comments and Suggestions for Authors
The authors have answered the questions raised by reviewers point by point.
Comments on the Quality of English LanguageOk